# Methylenetetrahydrofolate Reductase Polymorphism (rs1801133) and the Risk of Hypertension among African Populations: A Narrative Synthesis of Literature

**DOI:** 10.3390/genes13040631

**Published:** 2022-04-01

**Authors:** Sihle E. Mabhida, Babu Muhamed, Jyoti R. Sharma, Teke Apalata, Sibusiso Nomatshila, Lawrence Mabasa, Mongi Benjeddou, Charity Masilela, Khanyisani Ziqubu, Samukelisiwe Shabalala, Rabia Johnson

**Affiliations:** 1Biomedical Research and Innovation Platform, South African Medical Research Council (SAMRC), Tygerberg 7505, South Africa; sihlemabhida@gmail.com (S.E.M.); jyoti.sharma@mrc.ac.za (J.R.S.); lawrence.Mabasa@mrc.ac.za (L.M.); samukelisiwe.shabalala@mrc.ac.za (S.S.); 2Department of Biotechnology, Faculty of Natural Science, University of the Western Cape, Private Bag X17, Bellville, Cape Town 7535, South Africa; mbenjeddou@uwc.ac.za; 3Division of Infections Disease, University of Tennessee Health Sciences Center (UTHSC), Memphis, TN 38163, USA; bmuhamed2@childrensnational.org; 4Division of Medical Microbiology, Department of Laboratory-Medicine and Pathology, Faculty of Health Sciences, Walter Sisulu University, Mthatha 5100, South Africa; ruffinapalata@gmail.com; 5National Health Laboratory Services, Mthatha 5100, South Africa; 6Division of Preventive Medicine and Health Behavior, Department of Public Health, Faculty of Health Sciences, Walter Sisulu University, Mthatha 5100, South Africa; sibusiso.nomatshila@gmail.com; 7Department of Biochemistry, North-West University, Mafikeng Campus, Mmabatho 2735, South Africa; chemasilela@gmail.com (C.M.); ziqubukhanyisani@gmail.com (K.Z.); 8Centre for Cardio-Metabolic Research in Africa, Division of Medical Physiology, Faculty of Medicine and Health Sciences, Stellenbosch University, Tygerberg 7505, South Africa

**Keywords:** Hypertension, methylenetetrahydrofolate reductase gene, *MTHFR*, single-nucleotide polymorphism, Africa, genetic variation

## Abstract

In this review, we have gathered and analyzed the available genetic evidence on the association between the methylenetetrahydrofolate reductase gene (*MTHFR*), rs1801133 and the risk of Hypertension (HTN) in African populations, which was further compared to the global data evidence. This review was reported following the Preferred Reporting Items for Systematic Reviews and Meta-Analyses (PRISMA) protocol and Human Genome Epidemiology Network (HuGENet) guidelines. Literature was retrieved through major search databases, including PubMed, Scopus, Web of Science, and African Journal Online. We identified 64 potential studies, of which 4 studies were from the African continent and 60 studies were reported globally. Among the studies conducted in Africa, only two (*n* = 2) reported a significant association between the *MTHFR* (rs1801133) and the risk of developing HTN. Only one (*n* = 1) study population was purely composed of black Africans, while others were of other ethnicities. Among studies conducted in other continents (*n* = 60), forty-seven (*n* = 47) studies reported a positive association between *MTHFR* (rs1801133) and the risk of developing HTN, whereas the remaining studies (*n* = 14) did not show a significant association. Available literature suggests an apparent association between rs1801133 and HTN in global regions; however, such information is still scarce in Africa, especially in the black African population.

## 1. Introduction

Hypertension (HTN) remains a major risk factor for the development of cardiovascular diseases (CVDs), which significantly contributes to high rates of mortality and morbidity worldwide. Globally, HTN affects over 1.4 billion individuals above the age of 18 years and the number is expected to increase to 1.56 billion by 2025 [1,2,3]. In Africa, HTN affects approximately 74.4 million individuals [4,5]. Although there are various treatments available for HTN, it is apparent that patients are now gaining resistance to the treatment, and more severe cases have been recorded, particularly among individuals of African origin [5,6]. Furthermore, the high prevalence and severity of HTN that has been observed across different populations have been attributed to genetic variation [7]. Generally, it has been reported that genetic factors contribute to approximately 30–60% of the blood pressure (BP) variation that has been observed [8,9]. Therefore, it is critical to explore genetic factors with regards to HTN with the aim of understanding their role in the pathogenesis and progression of the disease.

Many approaches have been used to identify genetic variants associated with HTN in various populations [1,10,11]. The most common type of genetic variants are single nucleotide polymorphisms (SNPs), which represent approximately 90% of human genetic variations [12]. Several genome-wide association studies have identified multiple SNPs associated with HTN [13,14]. Amongst the predominantly identified variants is the SNP rs1801133 (position 677 C > T) found in exon 4 of the Methylenetetrahydrofolate reductase gene (*MTHFR*), which has been reported to be associated with elevated BP in various populations [15,16].

The *MTHFR* gene sits on the short arm of chromosome 1 (1p36.22), which has 12 exons and encodes for a protein containing 656 amino acids [17,18]. *MTHFR* is an enzyme that facilitates the production of 5-methyl-tetrahydrofolate, an active form of folate (Vitamin B9) in the body [19]. Previous research has demonstrated that 5-methyl-tetrahydrofolate is a positive allosteric modulator of nitric oxide synthase 3, which plays a significant role in the production of nitric oxide, a potent vasodilator in the regulation of BP [20]. Moreover, the *MTHFR* gene polymorphism has been suggested to be associated with increased levels of plasma homocysteine (hyperhomocysteinemia), which acts as an independent risk factor for HTN [21,22]. Factors like excessive alcohol consumption and smoking can influence the elevation of homocysteine in blood plasma [23,24]. According to a cross-sectional study conducted among women, the association between folate intake and homocysteine was altered by both alcohol intake and *MTHFR* rs1801133 [25].

A meta-analysis by Wu et al. [26] already demonstrated that *MTHFR* gene polymorphisms are linked with a significantly increased risk of HTN in subjects that carry the T allele and TT genotype. Another meta-analysis by Yang et al. [27], which was conducted in Indiana, United States of America, also reported an association between *MTHFR* (rs1801133) and HTN. However, this association was only significant among Asian and Caucasian populations, while no correlation was observed for Latinos, Africans, and Indians, suggesting the implication of ethnicity in disease susceptibility. Importantly, the authors acknowledged the essential limitations, such as the relatively small sample size and data scarcity for Latinos, Africans, and Indians. Furthermore, studies conducted in Morocco [16] and China [28] suggested that *MTHFR* (rs1801133) is associated with an increased risk of HTN. Conversely, Amrani-Midoun et al. [15], reported no association in the Algerian population. Thus, information regarding the correlation between *MTHFR* polymorphism (rs1801133) and HTN remains elusive, especially among black Africans. This review has extracted and critically analyzed the available clinical evidence on the association of *MTHFR* (rs1801133) and HTN in African populations, and further compared the evidence with studies conducted in other parts of the world.

## 2. Methods

### 2.1. Search Strategy

A comprehensive literature search was performed using subject headings or primary search (MeSH) terms such as “Methylenetetrahydrofolate reductase gene”, “*MTHFR*”, “hypertension”, “genetic*”, “single nucleotide polymorphism”, and “pharmacogenomics” (Appendix A) following the Human Genome Epidemiology Network (HuGENet) [29,30] and PRISMA guidelines [31,32] (Appendix A). The reference lists of included studies were further scanned for additional relevant studies. The search was done using major search engines and databases, including PubMed, Scopus, Web of Science, and African Journal Online. However, this review was not registered with online registries; therefore, the protocol does not have a registration number. Nevertheless, the aforementioned search engines and databases were thoroughly searched to make sure no other similar studies are currently underway.

### 2.2. Inclusion Criteria and Data Extraction

Studies included in the current review meet the following requests: (a) only the case-control studies were considered; (b) evaluated the *MTHFR* gene, rs1801133 polymorphism, and HTN risk; and (c) studies with data on the genotypes among cases and controls [33]. Studies were excluded if (a) conducted before the inception of molecular biology techniques (1983), (b) non-human studies, (c) family studies, and (d) reviews (Table 1). The data were independently and carefully assessed for compliance with the inclusion or exclusion criteria by three authors (S.E.M, K.Z, and C.M) who resolved disagreements and reached a consistent decision with the help of a fourth investigator (B.M). The following information was extracted from each study: the first author, publication year, country, ethnicity, continent, number of cases and controls, source of controls, and Hardy-Weinberg Equilibrium (HWE). Language restriction was applied during the search meaning studies conducted in other languages that could not translate into English were excluded.

## 3. Results

There are very limited studies reporting on *MTHFR* (rs1801133) association and HTN in African populations. Out of all identified relevant studies, only one study population was indigenous African (Cameroon) [34]; others were composed of Caucasian participants (Algeria, Morocco, Egypt) [15,16,35]. For this reason, a narrative synthesis of the findings was performed, instead of a meta-analysis.

### 3.1. Characteristics of Studies

Using our search strategy (Figure 1), we have identified 1230 related studies, of which four (*n* = 4) were from the African continent and (*n* = 60) were from non-African continents (globally). Based on our inclusion and exclusion criteria, there were 321 cases and 308 controls for the African population and 15,865 cases and 28,762 controls for other global populations globally, that were available for this analysis. The study characteristics are described in Table 1. In all the studies reported in Africa [15,16,34,35], HTN was defined as systolic/diastolic BP (SBP/DBP) ≥ 140/≥ 90 mm Hg. Among the included studies reported in the African region (*n* = 4), only two (*n* = 2) studies reported a significant association between the *MTHFR* (rs1801133) and the risk of developing HTN [16,34]. All African studies, where age was reported, included only patients aged above 40 years, except for Amin et al. [35], which included patients aged ≤ 45 years. Furthermore, most African studies included more females than males with exception of Amin et al. [35] which did not report on gender. In studies reported in other continents (*n* = 60), forty-seven separate studies showed a significant association between *MTHFR* (rs1801133) and the risk to develop HTN Table 2, whereas the remaining studies did not show any significant association (*n* = 14).

### 3.2. Association of MTHFR (rs1801133) and HTN Reported in African Continent

In this section, we briefly summarize the evidence on *MTHFR* (rs1801133) associations based on the four available studies reporting on African populations (Table 2).

The first study was conducted by Amin et al. [35], and it was aimed at evaluating the presence of *MTHFR* (rs1801133) polymorphism and its association with HTN and myocardial infarction among participants of Egyptian origin (*n* = 181, <45 and ≥45 years). The study showed that there was no association between *MTHFR* (rs1801133) and HTN. The study further demonstrated that individuals with HTN were smokers and presented with impaired lipid profiles such as significantly raised levels of total cholesterol (TC), triglycerides, low-density lipoprotein-cholesterol (LDL-c), and low high-density lipoprotein cholesterol (HDL-c), in comparison to the control group. The gender of the participants was not reported in this study. The authors clearly stated the guidelines (SBP/DBP ≥ 140/90 mm Hg) that were used to define HTN. However, the method used to adjust for patients who were already on treatment was not mentioned.

The second study by Nassereddine et al. [16] was carried out to evaluate the association between *MTHFR* (rs1801133) variant and HTN in a Moroccan population (*n* = 203, range 40–87 years). The authors demonstrated a significant association between rs1801133 and HTN. It was further demonstrated that the distribution of demographic and clinical characteristics of patients did not show a significant trend in relation to HTN. Thus, the study did not adjust for confounding factors. The study reported more females (*n* = 77) than males (*n* = 24). Lastly, the study defined HTN as SBP/DBP ≥ 140/90 mm Hg. However, the authors did not provide any information about the treatment status of the cohort. 

The third study by Amrani-Midoun et al. [15] reported a lack of association between *MTHFR* (rs1801133) and HTN in an Algerian population (*n* = 154, ≥42 years); however, the authors did acknowledge the impact of the small sample size used. Despite the small sample used, this study showed that there were significant differences between participants with HTN and controls with respect to age, SBP, DBP, and family history of HTN. The study was composed of more females (*n* = 84) than males (*n* = 70), and defined HTN as SBP/DBP ≥ 140/90 mm Hg. However, the method that was used for adjusting for the use of antihypertensive medication was not mentioned. Also, the genotyping method used in this study (PCR-RFLP) could be a potential limitation.

The fourth study was conducted by Ghogomu et al. [34], and it reported an association between *MTHFR* (rs1801133) and HTN in the native Bantu ethnic group of the South-West region of Cameroon (*n* = 91, range 40–70 years). Of note, this was the only study that sampled participants from an indigenous African population. Lipid profile dispersion for all subjects reported that serum lipid levels were higher in hypertensive patients than in healthy controls. The study further demonstrated that the *MTHFR* (rs1801133) variant may influence individual susceptibility to HTN through a mechanism that involves an increase in the level of serum LDL-c. However, the sample size was very small and was likely accompanied by biasedness. Furthermore, the study did not report on the number of females/males that were sampled. HTN was defined as having elevated SBP ≥ 140 mm Hg and DBP of at least ≥90 mm Hg. Patients who were already placed on hypertensive medication were also categorized as hypertensive.

## 4. Discussion

The *MTHFR* gene has been among the most studied genes associated with the development and progression of HTN [26,36]. Indeed, numerous genetic studies have investigated the association between the genetic variant of *MTHFR* (rs1801133) and the risk of developing HTN [36,37,38]. However, these studies reported conflicting results. In our previous systematic review, the *MTHFR* gene (rs1801133) was reported as one of the most studied genes associated with HTN among African populations [95]. Thus, in the present review, we gathered and analyzed the available genetic evidence on the association between *MTHFR* (rs1801133) and HTN among Africans and further compared the evidence with global data.

We reviewed 60 published articles that examined the association between *MTHFR* (rs1801133) and HTN. Out of 60 published articles, 47 reported a positive association between HTN and the *MTHFR* variant. However, only 4 studies were conducted in the African continent, of which 2 reported a positive association between rs1801133 and HTN [16,34]. The inconsistencies observed between these studies may be due to: (a) the limited number of relevant African studies and their relatively small sample sizes, which makes comparisons with other studies challenging. Given the small sample size in these studies, many true associations with small effects will not be significant and many suggestive associations may be false. In addition, the use of various cohorts, to maximize sample size and increase statistical power, could interfere with the biased results as some associations may be due to heterogeneity [96]; (b) the low frequency of the *MTHFR* (rs1801133) T allele observed among the African populations [97], which may be influenced by folate deficiency due to malnutrition and impaired intestinal absorption of folic acid, which are common in Africa [98]. Lastly, a study by Amrani-Midoun et al. [15] also suggested that these differences may be due to the epigenetic mechanisms which are involved in the gene expression predisposed by environmental factors such as lifestyle and diet. All the aforementioned factors may lead to failure to replicate the association of *MTHFR* (rs1801133) with disease phenotypes.

Although all included African studies [15,16,34,35] defined HTN as SBP/DBP ≥ 140/90 mm Hg, there were great differences in these studies, partly because of the criteria used in selecting participants and methods applied in each study. A study by Ghogomu et al. [34] and Nassereddine et al. [16] reported an association between *MTHFR* (rs1801133) polymorphism and HTN. However, these studies did not adjust for confounding factors such as gender, age, and smoking status. This may introduce bias, thus making it difficult to compare the findings with other studies. Furthermore, the age inconsistencies among the four African studies [15,16,34,35] may impose challenges when comparisons are made with other studies. For instance, a study by Nassereddine et al. [16], included 101 outpatients with a mean age of 61.6 ± 9 (range 40–87 years) and 102 age and sex-matched unrelated healthy control subjects with a mean age of 59.24 ± 10.7 (range 40–87 years); whereas a study that was conducted by Amin et al. [35] sampled young adults aged <45 years and older adults aged ≥45 years. The use of antihypertensive medication was reported by African studies [15,16,34,35]; however, the methods used for adjustments were not mentioned. This may introduce bias when making comparisons across studies, as studies that make adjustments would not be comparable to studies that did not make adjustments. Ghogomu et al. [34] was the only study that was composed of participants from an indigenous African population [15,16,35], thus limiting comparisons across different racial groups, since the genetics of HTN vary across different populations and geographical regions [17,99].

Nonetheless, a recent systematic review and meta-analysis comprising of 57 studies with 14,378 patients and 25,795 control subjects examined the association between *MTHFR* (rs1801133) and HTN and revealed that the major reason for equivocal results might be the racial differences observed across the different studies [36]. In comparison with that study, the present review had the following advantages: First, there were 64 eligible studies with 16,186 hypertensive cases and 29,070 controls, which could provide more reliable conclusions. Second, since none of the previous systematic reviews and meta-analyses [26,27,100,101,102] focused on the indigenous African populations, we assessed the comparison among studies reported on other cohorts with the ones reported on African populations. Therefore, future studies should pay more attention to the differences in the genetic background of indigenous African populations. For this reason, our review updates information from the previous systematic reviews [26,27,100,101,102] with additional supplements and adjustments, which makes it a comprehensive study regarding the association between *MTHFR* (rs1801133) and HTN.

### Strengths and Limitations of This Study

It must be pointed out that this is the first review that specifically assessed the effect of *MTHFR* (rs1801133) on HTN in Africa, as well as performing comparisons between African studies and available global data, which opens the door for future research. However, it should be noted that there were certain limitations to the present analysis, which inevitably prevented more in-depth analyses. First, the sample size of some of the selected studies was relatively small. Second, variations between population characteristics, phenotypic measures, and genotypic analyses could cause bias when comparing the current findings with previous reports. Third, literature was surveyed globally; unfortunately, in Africa, we were able to identify only four studies, which suggests that there is a lack of information regarding the black African ethnic groups in relation to genetic association studies. Furthermore, out of those identified studies (*n* = 4), only one study by Ghogomu et al. [34] was composed of a purely black African population and the remaining three studies were composed of other ethnicities [15,16,35]. Thus, our meta-analysis only included a few numbers of participants who were of African origin. As such, the analysis was unlikely to produce valid results (Figure 2), thus we conducted a narrative synthesis of the results. This indicates that there is an urgent need to carefully plan African-specific studies with large sample sizes in order to be able to draw conclusions on the association between *MTHFR* (rs1801133) and HTN.

## 5. Conclusions

Although the association between rs1801133 and HTN was predominantly reported in other global regions, the result from the current review opened avenues to further explore a possible association between rs1801133 and HTN among individuals of African origin. Furthermore, this study has demonstrated the need to generate African-specific genomic data. Such data could provide insights into human evolution and the role of genetic variants in disease phenotypes. These data could also increase our understanding of African population genomics and highlight its potential impact on biomedical research and genetic susceptibility to disease. Thus, future studies should sample a fair number of participants that completely represent the African population. Since African populations are well known to have high genetic diversity, because of their deep evolutionary history, and genetic differences, it is of utmost importance for future association studies to pay more attention to African genetic studies and to understand the functional and biological relevance of associated rs1801133. Moreover, improved methods need to be developed to understand and compare heritability across populations and study participants from different parts of the African continent. It is also imperative for all studies to report more detail in the protocols used to enable better replication and minimize bias between studies.

## Figures and Tables

**Figure 1 genes-13-00631-f001:**
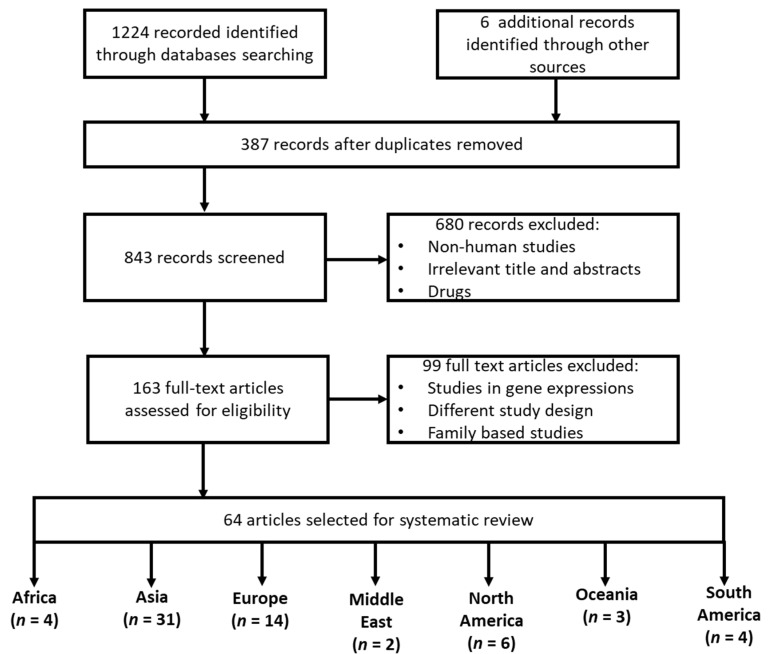
A flow-diagram showing an overview of study identification, inclusion, and exclusion criteria.

**Figure 2 genes-13-00631-f002:**
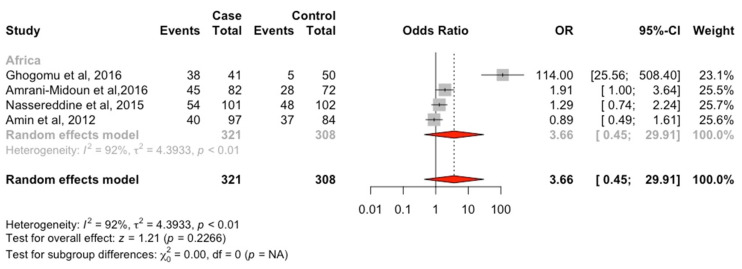
Forest plot of the evaluation for the association between the *MTHFR* (rs1801133) and HTN in the dominant genetic model (Africa). We evaluated the risk of the TT or CT genotype on HTN compared with the CC genotypes. Then, pooled Odds ratios (OR) with 95% confidence intervals (CI) and z score were performed to estimate associations. All analyses were performed using R software (Version 3.3.3, using R package meta) [15,16,34,35,103].

**Table 1 genes-13-00631-t001:** Inclusion Criteria and Data Extraction.

Inclusion	Exclusion
Published from 1984 to 2021	Studies conducted before 1983
Human studies	Non-human studies
Reported data on the genotypes among cases and controls	No genotypes among cases and controls
Studies reporting association between *MTHFR* polymorphisms (rs1801133) and HTN	Studies in gene expression
Studies provided enough data to calculate ORs and 95% confidence interval	Studies provided not enough data to calculate ORs and 95% confidence interval
Case-control design	Reviews
Non-family-based studies	Family-based studies

**Table 2 genes-13-00631-t002:** Main characteristics of studies included in this review.

Author, Year	Association	Country	Ethnicity	Cases	Cases with SNP	Control	Controls with SNP	*p*-Value HWE
**Africa**								
Ghogomu et al., 2016 [34]	Yes	Cameroon	Bantu	41	38	50	5	Yes
Amrani-Midoun et al., 2016 [15,36]	No	Algeria	Caucasian	82	45	72	28	Yes
Nassereddine et al., 2015 [16]	Yes	Morocco	Caucasian	101	54	102	48	Yes
Amin et al., 2012 [35]	No	Egypt	Caucasian	97	40	84	37	Yes
**Asia**								
Arina et al., 2019 [37]	Yes	Indonesia	Asian	53	21	53	10	Yes
Dwivedi et al., 2017 [38]	Yes	India	Asian	100	29	223	39	No
Fan et al., 2016 [28]	Yes	China	Chinese	214	177	494	375	Yes
Wen et al., 2015 [39]	Yes	China	Asian	174	129	634	376	Yes
Wang et al., 2015 [40]	Yes	China	Asian	190	94	287	143	Yes
Cai et al., 2014 [41]	Yes	China	Chinese	200	161	200	139	Yes
Xi et al., 2013 [42]	Yes	China	Chinese	619	378	2458	1376	Yes
Zhang et al., 2012 [43]	No	China	Asian	189	61	165	48	Yes
Cao et al., 2012 [44]	Yes	China	Asian	223	158	147	98	Yes
Yin et al., 2012 [45]	Yes	China	Asian	670	426	682	360	No
Liu et al., 2011 [46]	No	China	Asian	155	97	140	66	No
Cai et al., 2009 [47]	Yes	China	Chinese	130	53	39	8	Yes
Lin et al., 2008 [48]	Yes	China	Asian	50	31	123	50	Yes
Luo et al., 2008 [49]	Yes	China	Asian	442	182	195	57	Yes
Tang et al., 2007 [50]	Yes	China	Asian	252	113	195	57	Yes
Markan et al., 2007 [51]	Yes	India	Asian	153	48	133	28	Yes
Hui et al., 2007 [52]	No	Japan	Asian	261	178	271	167	Yes
Xing et al., 2007 [53]	Yes	China	Asian	695	493	509	327	No
Li et al., 2006 [54]	No	China	Asian	26	8	30	9	Yes
Hu et al., 2006 [55]	No	China	Asian	110	55	115	54	Yes
Kalita et al., 2006 [56]	Yes	India	Asian	28	10	32	11	Yes
Lwin et al., 2006 [57]	No	Japan	Asian	116	77	219	155	Yes
Liu et al., 2005 [58]	Yes	China	Asian	100	71	100	69	Yes
Sun et al., 2003 [59]	Yes	China	Asian	55	49	46	32	Yes
Wang et al., 2002 [60]	Yes	China	Asian	105	88	46	32	Yes
Zhan et al., 2000 [61]	No	China	Asian	127	83	170	108	Yes
Kobashi et al., 2000 [62]	Yes	Japan	Asian	184	120	215	132	Yes
Gao et al., 1999 [63]	Yes	China	Asian	127	83	170	108	Yes
Nakata et al., 1998 [64]	No	Japan	Asian	173	110	184	119	Yes
Nishio et al., 1996 [65]	No	Japan	Asian	47	31	82	53	Yes
**Europe**								
Bayramoglu et al., 2015 [66]	Yes	Turkey	White	125	60	99	43	Yes
Husemoen et al., 2014 [67]	Yes	Denmark	White	4694	2463	7697	3907	Yes
Ilhan et al., 2008 [68]	Yes	Turkey	Turk	78	42	100	28	Yes
Marinho et al., 2007 [69]	Yes	Portugal	Portuguese	64	49	128	71	Yes
Nagy et al., 2007 [70]	Yes	Hungary	White	101	52	73	41	Yes
Demir et al., 2006 [71]	Yes	Turkey	White	100	67	102	59	Yes
Cesari et al., 2005 [72]	Yes	Italy	White	90	50	90	48	Yes
Tylicki et al., 2005 [73]	No	Austria/Poland	White	90	50	90	48	Yes
Yilmaz et al., 2004 [74]	Yes	Turkey	White	64	35	47	23	Yes
Frederiksen et al., 2004 [75]	Yes	Denmark	White	1267	691	7971	4120	Yes
Rodriguez-Esparragon et al., 2003 [76]	No	Spain	White	232	149	215	120	Yes
Kahleova et al., 2002 [77]	Yes	Czech Republic	White	164	82	173	87	Yes
Benes et al., 2001 [78]	No	Czech Republic	White	193	120	209	123	No
Zusterzeel et al., 2000 [79]	Yes	Netherlands	White	76	44	403	198	Yes
**Middle East**								
Alghasham et al., 2012 [80]	Yes	Saudi Arabia	Qassim	123	50	250	65	Yes
Fakhrzadeh et al., 2009 [81]	Yes	Iran	Asian	160	61	76	40	Yes
**North America**								
Perez-Razo et al., 2015 [82]	Yes	Mexico	Mexican	569	423	590	465	Yes
Vazquez-Alaniz et al., 2014 [83]	Yes	Mexico	Mixed	194	132	194	140	Yes
Deshmukh et al., 2009 [84]	Yes	United States	White	42	20	118	66	Yes
Canto et al., 2008 [85]	Yes	Mexico	White	125	89	274	213	Yes
Rajkovic et al., 2000 [86]	Yes	United States	American	171	29	183	32	Yes
Powers et al., 1999 [87]	Yes	United States	American	122	76	114	60	Yes
**Oceania**								
Fowdar et al., 2012 [88]	No	Australia	White	377	207	393	218	Yes
Ng et al., 2009 [89]	Yes	Australia	White	38	24	80	40	Yes
Heux et al., 2004 [90]	Yes	New Zealand	White	247	160	249	144	Yes
**South America**								
Rios et al., 2017 [91]	Yes	Brazil	American	96	83	85	65	Yes
Fridman et al., 2013 [92]	Yes	Argentina	White	75	46	150	79	Yes
Fridman et al., 2008 [93]	No	Argentina	White	40	25	86	47	Yes
Soares et al., 2008 [94]	Yes	Brazil	American	30	17	16	7	Yes

## Data Availability

Not applicable.

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
