# Peer review of "Methylenetetrahydrofolate Reductase Polymorphism (rs1801133) and the Risk of Hypertension among African Populations: A Narrative Synthesis of Literature"

_genes, 2022, doi:10.3390/genes13040631_

Round 1
Reviewer 1 Report
The work is done carefully, but there are some types errors: line 63, 66, 87………
Reviewer 2 Report
Mabhida et al. have submitted a meta-analysis analyzing the role of methylenetetrahydrofolate reductase polymorphism, rs180133, in hypertension in the African population. Multiple prior studies have looked at this relationship in other populations, including prominent meta-analyses mentioned in this article’s discussion. These polymorphisms have been shown to play a significant role in hypertension in Latinos, east Asians and Caucasians. It should be noted that previously described population prevalence of the risk alleles is low in Black Africans and hence, prior studies have not had significant power to prove/disprove this association among Black Africans.
In this setting the authors set out to evaluate this relationship in the African population and compared it to a global population. Despite using a reasonable search strategy, they found only 4 small studies; and only 1 included Black African patients. I offer several suggestions and queries to the authors:
- I would caution against making the main point of the study the meta-analysis. Calling this study an evaluation of the “risk of hypertension among the African population” when only 1 of 4 studies included Black Africans has limited validity. I do not think the analysis supports the title. A focus on the meta-analysis is also limiting their discussion. A better study design would be a systematic review to review the 4 studies at length: look at their strengths/ weaknesses, definitions of hypertension and chart the direction future studies designed in the African population should take. Maybe even consider a power calculation given the lower prevalence of these risk alleles in the Black African population.
- Dividing the studies by continent to provide a global perspective is somewhat artificial. Are African Caucasians different than other Caucasian populations genetically? Why should they be analyzed separately from their European counterparts?
- Rather than concentrating on the statistical significance of the meta-analysis, a systematic analysis would stress the lack of statistical power and data in the Black African population and should be a call for further studies. Given the limited data available, the statistical significance and results of the meta-analysis carry very little significance.
- Please comment on the definitions of hypertension used in various studies and consider only including studies using a standardized blood pressure- based definition.
- Statistical analysis states that an Egger’s linear regression analysis was performed but the results are not presented. The funnel plot in Figure 4A does not appear symmetrical indicating possible publication bias. Moreover, the legend incorrectly refers to the funnel plot as a forest plot
Round 2
Reviewer 2 Report
The authors have made several changes to the manuscript. I believe these have enhanced the quality of the manuscript and several of my concerns have been addressed. I have a few suggestions to fine tune this manuscript:
- Throughout the introduction and abstract: Please review sentence structure and grammar. Certain re-formatted sections can be improved. Example: last sentence of introduction.
- Abstract: Conclusion section- Review the conclusion. English grammar/ sentence structure can be improved.
- Figure 1 needs to be updated to indicate the changed study design rather than meta-analysis
- Section 3.2: Results and discussion are mixed. The results should be presented in this section. The comparison to previously conducted studies can be moved to the discussion. This will bolster the discussion. The comparison to prior study findings can also be more robust.
- Limitations: Fourth limitation’s end should be re-worded: “... a meta-analysis was not possible. Any meta-analysis on the studies in African populations is limited by high heterogeneity and carries little significance given the low sample size and lack of indigenous African population representation.”.
Author Response
The authors have made several changes to the manuscript. I believe these have enhanced the quality of the manuscript and several of my concerns have been addressed. I have a few suggestions to fine tune this manuscript:
We appreciate the positive response and critical points raised by the reviewer. Please note, we have now addressed the suggested minor revisions.
- Throughout the introduction and abstract: Please review sentence structure and grammar. Certain re-formatted sections can be improved. Example: last sentence of introduction.
Response: We would like to thank the reviewer for this valuable comment, the introduction and abstract has been revised as suggested. All the changes on the manuscript are highlighted in red color.
Abstract: Conclusion section- Review the conclusion. English grammar/ sentence structure can be improved.
Response: Authors thank and acknowledge the reviewer’s comment the conclusion has been revised as suggested. All the changes on the manuscript are highlighted in red color.
- Figure 1 needs to be updated to indicate the changed study design rather than meta-analysis
Response: Thank you for your valuable comment, authors have now updated the figure 1 as suggested.
- Section 3.2: Results and discussion are mixed. The results should be presented in this section. The comparison to previously conducted studies can be moved to the discussion. This will bolster the discussion. The comparison to prior study findings can also be more robust.
Response: thank you for this valuable comment, the results and discussion has been revised as suggested. (Page 6-7; line 142-186). All the changes on the manuscript are highlighted in red color.
- Limitations: Fourth limitation’s end should be re-worded: “... a meta-analysis was not possible. Any meta-analysis on the studies in African populations is limited by high heterogeneity and carries little significance given the low sample size and lack of indigenous African population representation.”.
Response: Authors thank and acknowledge the reviewer’s comment, the sentence has revised now read as follows:
“Furthermore, our meta-analysis only included a few numbers of participants who were of African origin. Therefore, the analysis was unlikely to produce valid result (Figure 2). Thus, a narrative synthesis of the results was performed. This indicates that there is an urgent need to carefully plan African specific studies with large sample sizes in order to be able to draw conclusion on the association of MTHFR (rs1801133) and HTN.” (Page 9; line 261-265). All the changes on the manuscript are highlighted in red color.
